# Mechanisms, Techniques and Devices of Airborne Virus Detection: A Review

**DOI:** 10.3390/ijerph20085471

**Published:** 2023-04-11

**Authors:** Yuqing Chang, Yuqian Wang, Wen Li, Zewen Wei, Shichuan Tang, Rui Chen

**Affiliations:** 1Beijing Key Laboratory of Occupational Safety and Health, Institute of Urban Safety and Environmental Science, Beijing Academy of Science and Technology, Beijing 100054, China; m18813070133@163.com (Y.C.); wangyuqian1314@163.com (Y.W.); tsc3496@sina.com (S.T.); 2Department of Biomedical Engineering, School of Life Science, Beijing Institute of Technology, Beijing 100081, China; 3120201434@bit.edu.cn (W.L.); weizewen@bit.edu.cn (Z.W.)

**Keywords:** airborne virus, particulate matters, COVID-19, building ventilation, aerosol

## Abstract

Airborne viruses, such as COVID-19, cause pandemics all over the world. Virus-containing particles produced by infected individuals are suspended in the air for extended periods, actually resulting in viral aerosols and the spread of infectious diseases. Aerosol collection and detection devices are essential for limiting the spread of airborne virus diseases. This review provides an overview of the primary mechanisms and enhancement techniques for collecting and detecting airborne viruses. Indoor virus detection strategies for scenarios with varying ventilations are also summarized based on the excellent performance of existing advanced comprehensive devices. This review provides guidance for the development of future aerosol detection devices and aids in the control of airborne transmission diseases, such as COVID-19, influenza and other airborne transmission viruses.

## 1. Introduction

The airborne transmission of pathogen is a notable feature of the ongoing global pandemic of coronavirus disease 2019 (COVID-19) [1]. It appears that other significant epidemics and pandemics have not involved airborne transmission. However, it is widely accepted that the airborne transmission mode is an important mode for respiratory viruses and likely drives community spread [2,3]. Importantly, the mode of exposure of viral aerosols is via inhalation [4]. Viral aerosol contained in human exhaled air ranges from 0.3 to 10 μm. However, some of the viral aerosols will shrink by evaporation after leaving the respiratory tract, and some will form large droplets by combining with other particles. These virus aerosols contribute to pathogen transmission through the air. Airborne transmission includes short-range aerosol transmission (<1 m) and long-range aerosol transmission. Short-range aerosol transmission is characterized by rapid deposition and affects a distance of less than 1 to 2 m, and long-range aerosol transmission is the inhalation of respirable aerosols with particle size less than 5 microns and more than 1 to 2 m from the infected person [5,6,7]. However, the risk of airborne transmission at close-range is generally always higher than that of long-range transmission [8]. These aspects will result in a large number of opportunistic infections, posing a significant challenge to epidemic prevention and control. Aside from COVID-19 in 2019, airborne transmission played a role in the spreads of other pandemics such as SARS pneumonia in 2003 and Influenza A (H1N1) in 2009 [9].

The mechanism by which epidemic diseases are transmitted via aerosol is that viral particles and/or some attached to the environmental particulate matters (PMs) remain active and suspended in the air for long periods until being contacted by humans and deposited on mucous membranes, where they reproduce to form infections [10,11]. According to studies, the exhalatory actions of virus carriers, such as breathing, talking, singing, shouting, coughing and sneezing, result in the production of a certain amount of viral aerosols [12,13]. Furthermore, people can shed lots of virus into their exhaled breath and do not need to be coughing to do so. There is great variation in exhaled breath viral shedding and temporal dynamics over the course of infection [14,15]. Work by Kristen Coleman and others shows that coughing and singing can increase the viral load emitted into aerosols from the respiratory tract [16]. The virus-related PMs have a wide range of particle sizes, from less than 100 nm to more than 1 mm [17]. Large aerosol PMs precipitate faster than small ones, but they are also more likely to impact and stick to other PMs and form droplets due to adsorption [6]. PMs within 10 μm can be suspended in the air for 5 min. Airflow in the room can make the PMs larger than 5 μm move away from the location where it is generated by the airflow. PMs smaller than 10 μm can enter the human chest cavity, and PMs smaller than 5 μm can reach the fine bronchi and alveoli [18]. While factors such as UV light, humidity and temperature inactivate viral PMs, the droplet-forming viral PMs reduce these effects to some extent and are thus more infectious than viral PMs alone [19,20]. Viral infectivity can remain for many hours [21,22]. The relationship between the length of period that the person is in the space where the infected person has been and the probability of infection is an important issue. The investigation of this issue has significant implications for the development of prevention, isolation and decontamination policies. However, more accurate qualitative or quantitative analyses rely on high-efficiency and sensitivity viral aerosol particle collection and detection device.

In the natural environment, the concentrations of viral PMs are typically very low [23]. The real-time monitoring and early detection of potential infectious bioaerosls can help meet the significant technical challenges posed by COVID-19. The qualitative or quantitative analysis of aerosol PMs in the environment necessitates the use of bioaerosol samplers capable of isolating and concentrating PMs from air samples, which can then be combined with other biochemical analytical techniques. A good example is the quantitative real-time polymerase chain reaction (qRT-PCR), which is widely used to detect viruses due to its great sensitivity and reliability. Certain comprehensive devices (CDs) are now available to enable real-time determination of parameters such as concentration, dispersion and composition of environment aerosol PMs [24].

The entire procedure for airborne virus detection is divided into three steps. The first step is to separate the PMs from the air and gather them into the solution. This is accomplished by converting the PMs into a hydrosol sample [25,26,27,28]. Up until now, there has been a lot of interest in the ability of aerosol to hydrosol (ATH) techniques to help with virus collection and enable downstream detection applications. The second step, called hydrosol to hydrosol (HTH), is to further concentrate and purify the detection target PMs in hydrosol. The third step is to transport the concentrated liquid to a subsequent analytical device, which is a detection technique, such as direct or indirect detection of virus components like nucleic acids or proteins. The development and application of such device is obviously critical to ensuring the biosafety of environmental monitoring and the scientific nature of ventilation in indoor spaces. Combining bioaerosol samplers with other analytical detection techniques to achieve real-time monitoring of airborne viruses is challenging [29].

Theoretical studies can aid in the development of devices that target bioaerosols in a certain particle size range, such as a well-designed high air flow-rate electrostatic sampler (HAFES) that can target the H1N1 (peak diameter: 95 nm), as well as HCoV-229E (109 nm) [24,30]. However, the viral particles in the real environment are usually not alone in the air but rather attach to other PMs to exist in the air with a wide particle size range. Chains of infection due to sequential presence in the same indoor environment are often frequently hard to track down and establish. Thus, broad-range collection, efficient purification and sensitive analysis of viral PMs in the environment are critical to resolving this issue. Furthermore, combining these three processes to create a CD that collects extremely low concentrations of viral aerosol and transfers them to a detection device with the appropriate limit of detection (LOD) in a small volume of liquid for subsequent detection would reduce the response time for outbreak prevention and control, the range of full nucleic acids and the cost of controlling outbreaks. This will provide significant technical support for the prevention of diseases transmitted by aerosols, such as COVID-19. Therefore, this paper takes a review at recent advances in the collection and detection of airborne pathogen viruses from this perspective, including the compounding of multiple mechanisms and principles in the collection process and the application of novel methods as well as the development of back-end molecular diagnostic techniques to provide ideas for collection-detection devices. 

## 2. Methods

The reports reviewed in this study were sourced from ISI Web of Science, Pubmed and Google Scholar. Two categories of keywords were used: (1) aerosol, particle, sampler, collector, virus and bioaerosol; (2) SARS-CoV-2, detection, nucleic acid detection, anti-gen/antibody detection, PCR, loop-mediated isothermal amplification (LAMP), clustered regularly interspaced short palindromic repeats (CRISPR), microfluid and chip. At least one keyword from each category was used in each search. Results show more than 150 publications are relevant to aerosol particulate matter collecting, sampling and detection. Significant publications are summarized in Table 1 and Table 2. First, the primary mechanisms of aerosol PM collection are reviewed. Second, various techniques to enhance collection efficiency are presented and organized into enhanced collection principles. Third, current detection methods for SARS-CoV-2 are briefly described, with an emphasis on detection in the field of microfluidics. Finally, an indoor environment biosafety monitoring program is proposed based on various principles combined with ventilation of the indoor environment. For future research and development of aerosol collection-enrichment-detection devices, research and application directions are provided.

## 3. Sampling Techniques and Devices of Airborne Viruses

### 3.1. Primary Mechanisms

The fundamental mechanism of aerosol particle collection is that the PMs are separated from the air by specific principles (inherent property of the PM or some force). These mechanisms or properties include but are not limited to size filtration, mass differences of components in aerosol, gravity, electrostatic force and centripetal force (Figure 1). Each of these collection principles has its applicable scenarios. The following is a brief description of each aerosol particle collection mechanism from the standpoint of aerosol flow within the device, as well as comments of each device.

#### 3.1.1. Mass Differences of Components in Aerosol

Impactor, impinger and cyclone samplers use mass differences to separate PMs from air. The particulate matter’s mass in aerosol varies. When the aerosol flow changes, the larger mass PMs settle on the collection plate due to inertia differences, while the smaller ones and remaining air are discharged from the outlet. 

##### Impactor

As illustrated in Figure 1A, for the Impactor fundamental principle, under the action of a fan or pump, aerosols are sucked or sprayed into a channel with baffles (collection plates); PMs are deposited on the collection plates by inertia; and the remaining aerosols are discharged from the outlet end with uncollected PMs.

This mechanism has been used successfully in a variety of contexts [33,60]. The flow rate, the pipe diameter of the channel and the size of the baffles can all be modified to meet the required target particle size for collection. Additionally, depending on the requirement, it offers single-stage or multi-stage processing options. The single-stage processing option can be directed to collect PMs of a certain particle size section, whereas the multi-stage processing option can collect a wide range of PMs throughout a broad spectrum. The detachable multi-stage device has the capacity to collect PMs within a certain target particle size range as needed. However, smaller particle sizes require extremely high velocities to collect the PMs. For example, the Stage 6 of Andersen 6-stage sampler requires accelerating the aerosol to 76.4 FT/SEC when collecting particles of 0.01 μm [61]. This makes the nozzle fabrication process and cost high which make high flow rate collection impossible and increase the manufacturing precision, the cost and the damages from clogging [59]. The environment aerosol PM, on the other hand, needs to be transported to the back-end analysis unit for following analysis after collection. The applicability of this principle to high-efficiency collection devices is limited, since the elution of PMs from the collection plate is a challenging procedure.

##### Cyclone

As shown in Figure 1B for Cyclone principle, aerosols are sucked or sprayed into the annular channel under the action of a fan. Centrifugal force deposits PMs on the inner wall of the annular channel, and the remaining aerosol is discharged from the exit end with the uncollected PMs.

To achieve high aerosol flow rate, the aerosol can be kept flowing at a high speed in the annular channel. However, experiments show that the decay of collection efficiency is more visible when the flow rate is high [29]. An important problem is how to seek out the appropriate relation between flow rate and efficiency. In addition, the elution of PMs from the inner wall is a challenging issue, making it difficult to transport samples to the detection device.

##### Impinger

As shown in Figure 1C for the Impinger’s mechanism, aerosol is rushed straight to a liquid surface under the action of a fan; PMs are rushed into and dispersed or dissolved in the liquid by inertia; and the remaining aerosol and uncollected PMs are discharged from the outlet end.

Direct collection into the liquid greatly improves the efficiency of transporting the sample to the back-end detection module. Similar to the Cyclone, the relation of flow rate and collection efficiency is the focus. Additionally, using multiple collecting fluids to handle various types of PMs with various properties, such as oily PMs, non-oily PMs and improving the collection fluid’s affinity for certain types of PMs, improves the collection efficiency of the PMs. However, re-aerosolize poses a serious concern due to the collection fluid is in a high flow rate aerosol environment, and the loss of collected PMs during the process is a challenge [62,63].

#### 3.1.2. Filtration

As shown in Figure 1D, the aerosol is passed through a channel containing a filter plate or membrane under the action of a fan. The PMs are trapped in the channel by the effects of the screen, and the remaining aerosol is discharged from the outlet end with the uncollected PMs. Filtration generally uses porous media like activated carbon, glass fiber nonwovens and medical stone as filter membranes [64,65,66]. The PMs in the aerosol are adsorbed in the pores inside the media when they pass through.

PMs larger than a specific particle size are collected using particular sieve membranes or screens (filtered). It is a good collection method if the target particle size is known. High flow rate collection is difficult to achieve when the filter pore size is designed for capturing nano or micron PMs. In addition, transporting of the PMs from the filter membrane to the back-end detection device requires a more laborious process, and the filter membrane itself is prone to clogging that prevents it from being utilized for an extended period.

#### 3.1.3. Electrostatic Precipitator

The electrostatic precipitator mechanism has been widely used in the field of dust removal [67,68]. Electrostatic precipitators are potential high-flow viral aerosol particulate collection devices because they can use Coulomb forces to all charged PMs in the aerosol to deflect and deposit them on the pole plate. The aerosol enters a collection electric field under the action of a fan, and the PMs are deflected and deposited on the collection plate by the Coulomb force. The remaining aerosol, along with the uncollected PMs, is discharged from the outlet. As shown in Figure 1E for electrostatic precipitator, the Coulombic force applied indiscriminately to all PMs. It provides the potential for high-efficiency collection. Additionally, similar to the Cyclone and Impinger, the relation between flow rate and collection efficiency is critical. Simultaneously, the electric field may have some effects on the activity of the PM due to the ozone produced by the negative electrode, which may influence the sensitivity of the culture and the subsequent analytical techniques [69,70,71]. However, certain techniques, such as targeting nuclear acid, may be unaffected.

#### 3.1.4. Particle Amplifier

High humidity causes each particle to swell with the water which condenses on the particles. Thus, each particle gets larger due to deliquescence/condensation and thus grows and becomes easier to be collected. The PMs are amplified and become easier to deposit before settling into the collection pool. As shown in Figure 1F, the low temperature aerosol enters a warm and extremely humid pipe under the action of a fan. In the pipe, the particles continuously grow to form larger droplets. The amplified droplets are deposited in the collection pool at the back end, and the remaining aerosol along with the uncollected PMs is discharged from the outlet end.

Particle amplifiers have the ability to achieve high collection efficiency while also meeting the needs of individual detection device. Multi-circulation of PMs can further improve their collection efficiency [72]. However, high flow rates are difficult to achieve due to the lengthy deposition process in the pipeline. Furthermore, the particle amplifier consumes quite a lot of energy.

#### 3.1.5. Application Examples

Various devices have been developed based on the fundamental mechanisms stated above. The commercialized devices have previously been reviewed [73,74,75,76,77,78]. Particle collection using only primary mechanisms has two troubles. First, the whole flow rate is low, and the collection efficiency decreases when the flow rate increases. Second, sampling particle size is limited, and the collection efficiency decreases when the particle size decreases. Even so, these primary mechanisms have been shown to be effective. Enhancement techniques, as well as the combination of several primary mechanisms, may have an extra positive impact on the high-flow efficient particle sampling.

Álvaro et al. used a PM impactor (air flowrate of 30 L/min) to sample the aerosol in a hospital with decontamination measures and obtained a positive sample in one hospital room [79]. Li et al. used a cyclone method sampler (Yao-CSpler) with the assistance of an intelligent robot to sample ambient aerosols from hospitals and isolation hotels and showed that the sampler was able to collect airborne bacteria, fungi and also viruses such as SARS-CoV-2 [80]. Li et al. sampled several public places using the cyclone method and 12 SARS-CoV-2 positive samples out of 23 samples [81]. Lane et al. used NIOSH BC 251 two-stage cyclone samplers to perform aerosol sampling in a dedicated hospital COVID-19 room and showed that 3% (19) of 576 aerosol samples collected from 19 different rooms of 32 participants were positive for SARSCoV-2, with mostly from near the head and foot of the bed [82]. Liu et al. used presterilized gelatin filters (Sartorius) and Sioutas Impactor (SKC) in two hospitals in Wuhan. The results showed very low concentrations of SARS-CoV-2 RNA in aerosols detected in isolation and ventilation wards but higher concentrations in the toilet areas used by patients [23]. Carlo et al. used a Sartorius Airscan microbiological sampler (Sartorius AG, Gottingen, Germany) with a gelatin membrane filter (80 mm diameter) at 50 L/min for sampling the isolation chamber. The direct release of SARS-CoV-2 RNA during normal breathing was studied in five patients using another microbiological sampler (portable Sartorius AirPort) with a gelatin membrane filter (2 mm diameter) at a flow rate of 80 L/min. The sampling time was 100 min. Viral RNA was detected in the air 2 cm from the mouth of patients who tested positive for SARS-CoV-2 RNA by oropharyngeal, nasopharyngeal and saliva swabs. In contrast, no viral RNA was identified in the exhaled air of patients who tested negative by oropharyngeal, nasopharyngeal and saliva swabs [83].

The performance of eight commercially available samplers was evaluated using SARS-CoV-2 containing aerosol particles by Ratnesar-Shumate et al. Biosamplers, PTFE filters, Sioutas impingers, gelatin filters and BC 251 samplers were all able to measure similar concentrations of infectious SARS-CoV-2 in aerosol and are able to maintain virus infectivity over the course of a 33 min sampling period. Both the all-glass impinger and the two small impingers showed significant loss of infectious virus over both 13 and 33 min of sampling, suggesting that these samplers may not be suitable for accurately sampling infectious airborne SARS-CoV-2 over longer sampling times [76].

There is only one type of mechanism for each of the above samplers, and although it can present good sampling results, its sampling efficiency and flow rate, etc., are limited by the mechanism itself. Prolonged sampling will cause more accumulation of PMs in the liquid sample, and we tend to refer to this approach as collection. The activity of microorganisms in samples obtained by collection cannot be guaranteed, and such samples are more appropriately used for molecular diagnostics. Samples obtained by short time sampling can be incubated and then analyzed. Enhanced and composite techniques are described later. The devices to which these techniques have been applied are also displayed.

### 3.2. Enhancement Techniques

Researchers have used enhancement methods to increase the collection efficiency and developed comprehensive device to improve the sampler collection efficiency based on the primary methods. The collection of aerosol is divided into three steps as described above: aerosol entry, aerosol collection via a specific principle, and residual aerosol discharge. As an extension, the collection efficiency can be increased from various extra steps of enhancement techniques (Figure 2).

#### 3.2.1. Aerosol Precharging

In nature, viral particles have relatively weak negative charges [84,85,86]. The charge of the PMs directly determines the magnitude of the Coulomb force on them when using electrostatic precipitation methods to collect PMs. A series of theoretical and experimental studies have been conducted on the particle charging characteristics, and particle charging laws corresponding to various charging devices, particle size and ionic density have been obtained. The precharge on aerosol PMs increases the Coulomb force and shortens the length of the device’s collection section, enabling PMs to be deposited onto the collection plate faster (Figure 2A) [87,88,89,90]. Priyamvada discovered that the precharging technique increased the efficiency of PM collection from 16% to 49% [37]. Alternatively, the premixing of charged aerosols in the inlet section may serve as another potential charging method for increasing PM collection efficiency (as shown in Figure 2B).

#### 3.2.2. Liquefied Collection Plate

The PMs are collected by devices such as impactor, cyclone, precipitation, etc., into a solid collection plate. These methods have a number of limitations: (1)PMs have to be eluted into liquid before conducting further monitoring tests;(2)There is greater potential for viral infectious decay on dray plates. Additionally, this is also a challenge with most methods;(3)The high velocity of wind may cause PMs to re-aerosolize.

Researchers have developed liquefied collection plates to address these issues (Figure 2C,D). In order to improve the collection efficiency, it is possible to collect the viral PMs directly in the liquid and deliver them to the detection module at the back end by turning the collection plate of the above methods into a collection pool. It was also shown that liquefying and charging the collection plate for electrostatic aerosols can still generate an electric field with collection capability in verification experiments [24,29,38]. The liquefied collection plate can be connected with the subsequent detection module using an automated device, which will shorten the overall process time and facilitate the automatic results display. Liquid samples can also be obtained using novel elution methods. The electro-wetting-on-dielectric (EWOD) technique allows droplets on the substrate to be manipulated by applying an electrical signal to the flow of electro droplets, thereby increasing elution efficiency (Figure 2D) [91].

#### 3.2.3. Cavities and Ribs on the Inner Wall of the Pipe

Well-designed cavities and ribs on the inner wall of an aerosol channel can improve the collection efficiency of PMs by causing some of them to be deposited in cavities, and such studies are typically carried out using simulation methods. The mechanism of deposition enhancement and deposition efficiency ratio of PMs with different particle sizes, rib spacing (p) and rib height (e) was investigated [92]. The presence of ribs on the channel’s inner wall surface improves the particle deposition efficiency significantly, and the entrainment of turbulent eddies caused by ribs is better for the capture of PMs with smaller particle sizes. The smaller particle are refer to dimensionless particle relaxation time <1 [92]. The dimensionless particle relaxation time is directly proportional with the square of the particle diameter [92]. Particle deposition efficiency improves as rib spacing decreases, especially for large PMs, but rib height variation has no significant effect on deposition efficiency. The deposition enhancement was highest when the ratios of rib spacing to rib height p/e = 2. Lu et al. also simulated aerosol particle deposition in smooth and pipelines decorated with cavities and ribs [92]. It was found that the addition of cavities and ribs to the inner wall of the duct increased the particle deposition rate by 10–4000 times either in the vertical or horizontal conditions. The maximum particle deposition enhancement occurred for PMs of 0.2–3 μm in size, while the minimum enhancement occurred for PMs of 20–50 μm in size. The design of cavities and ribs is an effective enhancement method for small particle sizes.

Hemmati et al. used the Euler–Lagrange method to simulate granular gas flow for particle deposition in a turbulent incompressible gas flow in a horizontal channel with different two-dimensional artificial roughness shapes (such as triangles, rectangles and semi-ellipses) and heights [93]. The shape of the ribs was discovered to influence the size of the gas flow field and recirculation zone, which in turn influence particle deposition efficiency. Triangular shaped ribs were found to have higher PM collection efficiency than other designs [94,95]. These studies have shown that cavities and ribs on the inner walls of sampler pipes improve the PM collection efficiency to some extent, as shown in Figure 2E. The following factors have the greatest impact on the efficiency of cavities and ribs: (1) Ribs spacing, height ratio: m = p/h; (2) Ribs shape.

In addition to the studies mentioned above, several others have investigated the effect of cavities and ribs on particle deposition efficiency using computer simulations, which can be used to guide the design of rough surfaces as well as structural design of certain inner walls of particle collection devices [96,97,98,99].

#### 3.2.4. Hydrosol to Hydrosol Enrichment

Generally, the concentration of hydrosol samples after collection by aerosol collector is insufficient. Enhancing the concentration of viral aerosol into hydrosol samples with efficient enrichment can enhance the performance of CD and support downstream detection that requires small sample volume. Various HTH enrichment methods have been developed to meet this challenge. Yeh et al. developed a portable microfluidic platform using carbon nanotube arrays with differential filtration porosity for rapid enrichment of viruses [100]. Twist Bioscience has developed a variety of SARS-CoV-2 specific platforms that target fragments of viral DNA for enrichment [101]. Wylezich et al. developed a virionome based on the biotinylated RNA decoy principle for specific capture enrichment of endemic animal and zoonotic viruses (VirBaits) [102]. Isaac A.M. Frias et al. used layered composite polyaniline-(electrospun nanofiber) hydrogel mats (ENM) for simultaneous enrichment and impedance sensing of ZIKV virus particles for specific identification of ZIKV in Vero cell cultures [103]. Bai et al. developed a magnetic quantum dot nanobead (MQB)-based lateral flow assay for magnetic enrichment of influenza A virus (IAV) particles in clinical specimens. Their method enables quantitative detection of IAV viral particles in 35 min with a detection limit as low as 22 pfu/mL and shows good specificity between influenza B virus and two adenovirus strains [104]. The aptamer used for synthesis and modification is capable of specific binding to the target, has demonstrated its functionality in bio-detectors and is expected to be used for specific enrichment of aerosol detection systems [105,106,107].

#### 3.2.5. Other Issues

Enhancement from all CD steps is required to improve the performance of airborne virus collection devices. Other methods, in addition to the various enhancement techniques mentioned above, are listed below. As an example, the Ace glass impinger (AGI-30) has a very low collection efficiency (30%) for fine particulate collection, and the glass bead filling has increased the efficiency to 99% [108].

The operators’ safety is another critical issue in the development of airborne detection CDs. Some substances are specifically raised here in order to achieve inactivation through premixing in the inlet section (Figure 2B). Tea tree and eucalyptus oils were used as a coating for filter fibers, which effectively improved the particulate collection efficiency while reducing the activity of viral PMs [109,110]. Damit et al. developed an ultra-high temperature (UHT) system to inactivate the virus and avoid secondary infection of the operator due to the operation of the device [19]. Kalaiselvan et al. indicate that essential oil vapors of Melaleuca alternifolia act on aerosol coronaviruses, *E. coli* or fungal Aspergillus flavus spores to reduce the number of culturable microbial cells [111]. Schnitzler et al. investigated the inhibitory ability of Australian tea tree oil (TTO) and eucalyptus oil (EUO) against herpes simplex virus. TTO was found to significantly reduce viral titers, while EUO exhibited significant but weak antiviral activity, although their active anti-herpes components are not known. However, they have promising applications as antiviral agents in recurrent herpes infections [112]. Garozzo et al. investigated the mechanism of action of TTO against influenza A (H1N1) virus subtypes and found that TTO does not inhibit influenza virus neuraminidase activity but rather inhibits viral uncoating by interfering with the acidification of the lysosomal internal compartment [113]. Shao et al. showed that TTO has a blocking effect on vesicular stomatitis virus (VSV) replication at the gene and protein level. This offers potential for its use in the control of SARS-CoV-2 virus, and relevant experiments can be conducted to validate it [114]. Miyaoka et al. have demonstrated that spraying on aerosols with acidic hypochlorite water (SAHW) can lower viral activity and hence prevent infections [115]. Jung et al. developed a generation method for nano-silver PMs [116]. They disclosed that the nano-silver PMs inhibited the activity of viral PMs. Notably, the type of aerosol PMs mixed in can be adjusted to change its affinity for specific PMs. It is possible to collect a defined particle size interval while consuming less energy. Furthermore, this eliminates the waste caused by broad-spectrum collection.

### 3.3. Comparisons on Airborne Virus Sampling Devices

Devices for collecting aerosol PMs using only one collection method already perform well, but they are far from adequate for practical scenarios [117]. Real-time surveillance is critical for airborne infectious diseases and epidemics. Further advancements in performance may allow it to be applied to practical epidemic prevention. To achieve increased efficiency, new CDs have been developed that use a combination of multiple primary mechanisms or the method of enhancing techniques. The following are some advanced aerosol particulate collection devices. These devices are critical for the indoor airborne virus detection applications (Table 1).

#### 3.3.1. Combinations of Multiple Primary Mechanisms

Using a cyclone and impactor, Sung et al. designed a CD in which aerosols enter from the outer wall in a cyclonic manner impacting on the collection plate and exiting from the inner channel in the opposite direction. With a flow rate of 1000 L/min, the efficiency of the air sampler is about 50% for a particle size of 1 µm and 78.3% for a particle size of 1.5 µm. For particle size greater than 1.5 µm, collection efficiencies of approximately 100% were observed [31,32]. A three-stage high-capacity bioaerosol sampler was designed [33,34]. Size-selective sampling of bioaerosols was performed by inhaling air at a high flow rate of 1000 L/min. In stage 1 and stage 2, PM > 10 μm and PM between 2.5 and 10 μm were collected using the cyclone method, respectively. In stage 3, PMs < 2.5 μm were collected using the filtration method, and generic Phosphate buffered saline (PBS) elution was used to obtain a hydrosol sample. This device combines the primary mechanisms, cyclone and filtration, to expand the collection range of particle size spectrum and enhance the collection efficiency of each particle size spectrum under the condition of ultra-high flow rate. Hong et al. designed a personal electrostatic particle concentrator (EPC) using an impactor and an electrostatic precipitator. The collection efficiency was improved at a lower flow rate of the individual sampling device. At a flow rate of 1.2 L/min, the collection efficiency of polystyrene PMs with particle sizes of 0.05–2 μm was up to 99.3–99.8% [35]. McDevitt et al. designed and built a new sampler, Gesundheit II (G-II); using an amplifier, the water vapor in the air condenses on the particles to form large droplets, and later the amplified droplets (because of deliquescence and condensation) are collected using an impinger. This device provides a collection efficiency of >85% for PMs larger than 50 nm for viral aerosols [36].

#### 3.3.2. Applications of Enhancing Techniques

The following describes the CDs using the enhancing techniques in combination with the primary mechanisms.

Tan et al. designed a new hemispherical electric field using a Liquefied acquisition plate and a pre-charging, electrostatic precipitator and developed automated bioaerosol collection and output [29]. After the aerosol is charged by the charging electrode, it enters the hemispherical electric field, where the travel path of the PMs is shifted by the Coulomb force and falls into the central charged collection fluid, which is circulated by an external pipe and a peristaltic pump to achieve a continuous cycle and can be integrated with subsequent detection modules. Under the conditions of collection voltage 20 kV, charging voltage 1.5 V and flow rate 1.2 L/min flow rate, the collection efficiency is 70% to 90% for PMs of 0.3–1.2 μm, >90% for PMs of 0.65–0.8 μm and >90% for PMs of 0.8–2.0 μm; under the conditions of 6.2 L flow rate for each particle size segment, the collection efficiency of PM is reduced to 20–60% [29]. This device employs a hemispherical electric field and liquefied acquisition plates to efficiently collect PM into liquid samples. Kim et al. used a combination of a Liquefied acquisition plate and electrostatic precipitator [24]. A High air flow-rate electrostatic sampler (HAFES) was designed. The collection efficiencies were 88%, 79%, 82% and 71% for HCoV-229E particles with peak particle size of 109 nm at a collection voltage of −10 kV and fluxes of 40 L/min, 60 L/min, 80 L/min and 100 L/min, respectively. This device has a high collection efficiency, and it has been discovered that the collection efficiency decreases as the flow rate increases. Priyamvada et al. developed a new electrostatic precipitation-based portable low-cost sampler (TracB) using the principles of prefixing and electrostatic precipitator [37]. The collection efficiency is above 50% for PMs with particle size 0.01–10 μm at 8 kV and flow rate 10 L/min. The Liquefied acquisition plate, pre-charging and electrostatic precipitator principles were used by Ma et al. [38]. The integrated micropump automatically delivers the acquisition fluid through a half-open microchannel to the liquid outlet and flushes the deposited PMs away into the collector, which can be integrated with the detection module at the back end. For PMs < 5 μm, the maximum sampling flow rate is 13.2 L/min at a charging voltage of −1.8 kV and a collection voltage of −7 kV, with a corresponding maximum effective collection efficiency of about 40%. This integrated microfluidic electrostatic sampler (IMES) is a bioaerosol sampling system ready for integration with subsequent automatic detection device. Lin et al. developed a high-flow water-based condenser by adding an aerosol premixing link to the front end of the cyclone to amplify the PM and enhance the collection efficiency [39]. The results showed that when the aerosol flow rate was 19 °C and the temperature of the mixed reservoir was 50 °C, the physical capture efficiency of aerosol of 30–100 nm was 70–99%. This device makes better use of the amplifier to improve overall PM collection efficiency, but the amplified PM can result in inaccurate detection data at the back end. Novosselov et al. developed and tested a low-cost micro-channel collector (mCC) using a Micro-Channel cyclone [40]. PMs are collected on the wall of the pipe due to centrifugal force. The typical collection efficiency is more than 50% for 0.5 μm PMs and 90% for PMs larger than 1 μm. This device has potential applications for front-end microfluidic chip acquisition.

### 3.4. Differences between Virus Sampling and Virus Detection

It is critical to differentiate between sampling and collection. Because sampling-based devices do not require real-time data, the samples are subsequently re-cultured, and they require less efficiency. In less human-populated environments, such as farms and pastures, sampling equipment can be used for routine monitoring. Collection devices must be more efficient, as all PMs passing through the device must be collected for subsequent testing of certain critical components. Devices for data collection can be used in densely populated environments such as conference rooms, subway stations and large shopping malls. Based on the analysis and comparison in Table 1, the following classification criteria are proposed.
(1)High flow rate: >1000 L/min; medium flow rate: 100–1000 L/min; low flow rate: <100 L/min.(2)Particle size spectrum—collection efficiency: high collection efficiency ≥ 70%, medium collection efficiency: 40–70%, low collection efficiency: <30% low collection efficiency.(3)Collection efficiency—particle size spectrum: full particle size collection device, partial particle size collection device, targeted collection device.

## 4. Detection Techniques of Airborne Virus

Detection techniques are important for compounding devices and determine the amount of hydrosol sample per unit time, sensitivity and response time. Virus assays are usually divided into two main categories, namely direct and indirect assays [118]. Indirect detection methods involve virus isolation, where viruses are introduced into a suitable host cell line to propagate virus PMs, which are later tested. This method usually requires a longer turnaround time. Direct detection methods include nucleic acid detection and immunoassay. Nucleic acid amplification test (NAAT) refers to the amplification and detection of genes in the virus and is characterized by early diagnosis, high sensitivity and specificity, and the most widely used technique is RT-PCR. Immunoassay uses antibodies as the main means of detecting the virus in the sample, with the greatest advantage of convenience and short detection time. However, immunoassay detection of infection may be of limited use in the early stages due to the time required for the body to develop an immune response to the infection. Table 2 summarized the typical detection techniques. Current conventional diagnostic techniques require expensive device and specialized operators and are inadequate to enable rapid, accurate and on-site diagnosis during pandemics. 

### 4.1. Nucleic Acid Detection Techniques

Nucleic acid testing has a relatively high sensitivity. In general, nucleic acid testing is used to confirm a viral infection diagnosis. PCR, LAMP, CRISPR and other nucleic acid detection methods are commonly used. Current conventional diagnostic techniques require expensive devices and specialized operators and are insufficient for rapid, accurate and on-site diagnosis during pandemics. Microfluidic devices provide a favorable platform for rapid testing because they offer highly integrated disposable chips that enable potential automation from sample preparation to result output (Figure 3). Microfluidic techniques necessitate a certain concentration and purity of sample concentration, and after converting the virus aerosol into a liquid sample, ultracentrifugation or incubation is frequently required to enrich the virus particles or increase their titer. [119,120]. The HTH enrichment methods reviewed in Section 3.2.4 provide an idea for integrating microfluidic techniques with various collection methods. In general, virus concentrations in environment air are very low; e.g., SARS-CoV-2 concentrations of 0.87 virus genomes/L were found at a university student health care center in the study by John et al. [120]. SARS-CoV-2 concentrations of <3 copies/m^3^ or 16 to 42 copies/m^3^ were found at a hospital in the study by Liu et al. [23]. Therefore, a large flow rate of air is required to transfer enough samples to downstream analysis. Microfluidic technology has recently provided a breakthrough in viral detection. Several microfluidic-based nucleic acid detection platforms have been introduced.

#### 4.1.1. Polymerase Chain Reaction

Many studies on microfluidic systems related to PCR have been conducted as a hot topic in nucleic acid diagnostics. PCR is a thermal cycle nucleic acid temperature amplification technology. Lee et al. described a disposable, polymer-based RT-PCR detection chip for human immunodeficiency virus (HIV) testing [121]. The chip incorporates embedded micro pinch valves and RT-PCR to produce a portable analyzer with an infrared lamp temperature control system and an optical detection system for analytical chemiluminescence assays. The detection takes only 35 min, reducing contamination and sample loss. Oshiki et al. used microfluidic nested PCR followed by MiSeq amplicon sequencing (MFnPCR–MiSeq) to create a high-throughput tool for detecting and genotyping 11 human pathogenic RNA viruses [122]. This is the first study to use microfluidic PCR and next-generation sequencing technologies in a genotyped environment to detect and genotype multiple human RNA viruses. Li et al. developed an automated sample-to-answer disk based on a dual rotary axis centrifugal microfluidic platform to detect hepatitis B virus (HBV) in whole blood [123]. The disc can separate serum from whole blood, extracting DNA using magnetic beads and aliquot nucleic acids and performing real-time polymerization chain reactions. For HBV DNA detection, the sample-to-answer time is about 48 min; the detection limit is 100 copies/mL; and the only manual processing step is supplying the sample to the disc.

#### 4.1.2. Loop-Mediated Isothermal Amplification

Isothermal amplification requires few devices, can be done in a thermostatic water bath and has a quick reaction time. It can be highly integrated with microfluidic devices to achieve device miniaturization, high sensitivity and precision of nucleic acid detection. Ma et al. developed a self-driven microfluidic device that detects H1N1 influenza virus using reverse transcription loop-mediated isothermal amplification (RT-LAMP) [124]. A hydrophobic soft valve, novel polydimethylsiloxane surface treatment and capillary force were used to build the device, which performs virus isolation and lysis, isothermal nucleic acid amplification and virus colorimetric detection. The entire procedure only takes 40 min and has a detection limit of 0.8 ng/mL. It is adequate for use in an environmental detection setting. LAMP reaction can also be used for factual monitoring. The use of fluorescent dyes causes false positives due to non-specific amplification. It can be used to distinguish nonspecific amplification by labeled primers [125,126], binding quenching probes [127], hybeacons probes [128] and molecular beacons [129]. The future development trend is a fully integrated inspection system. There have been numerous reports on materials suitable for nucleic acid extraction, including modified magnetic beads, silica beads/sol gel solid-phase extraction (SPE) beds and silica/glass columns. Wang et al., for example, proposed a microfluidic system that integrates RNA extraction and LAMP detection processes for detecting nervous necrosis virus (NNV) in aquacultured grouper [130].

#### 4.1.3. Clustered Regularly Interspaced Short Palindromic Repeats

Due to its high sensitivity and specificity, mobility, quickness and low cost, CRISPR has been created as an effective instrument for nucleic acid detection [131,132,133,134,135]. The CRISPR approach is also suitable for bedside testing because it can be carried out at physiological temperatures. The creation of CRISPR-based biosensors as point-of-care (POC) devices is made possible by microfluidic technologies. The SARS-CoV-2 RNA can be concentrated and purified from a significant amount of swab eluate using a single-step CRISPR-Cas-assisted assay (POC-CRISPR), which uses magnetic-based capture and transport of nucleic acid-binding magnetic beads to transport the RNA to downstream amplification and detection [136]. POC-CRISPR can detect SARS-CoV-2 RNA—1 genome equivalent/μL from a sample volume of 100 μL less than 30 min. In an effort to speed up the CRISPR-Cas12 reaction, Ramachandran et al. used isoelectrophoresis (ITP) on a microfluidic chip to harvest RNA from nasopharyngeal swabs. The procedure includes isothermal reverse transcription and amplification, ITP-based nucleic acid extraction from the original sample and ITP-enhanced CRISPR testing [137]. With an average LOD of about 10 copies/μL and a sample to result time of only 40 min, this approach uses approximately 100 times less volume of all reagents than the average quantity of only 0.2 μL.

### 4.2. Antigen Detection Techniques

Antigen detection is to determine the presence or absence of a virus by detecting proteins on the surface or inside the virus. Here, we focus on several virus sensing platforms.

#### 4.2.1. Surface Plasmon Resonances

Surface plasmon resonances (SPRs) have been used to identify biomolecular interactions for therapeutic or scientific purposes. They can detect very low concentrations of chemical and biological substances near the measured surface by monitoring the region’s refractive index value in real time [138]. When a target analyte is captured by a bioreceptor immobilized on the sensing layer, a change in the sensor surface structure causes a change in refractive index. The use of different receptors allows the capture of different pathogens, which enables specific detection of pathogens [139,140]. For example, Yoo et al. reported a reusable magnetic SPR sensor chip in which they used ferromagnetic patterns fabricated on the SPR sensor chip to trap a layer of magnetic PMs, and those patterns with trapped magnetic PMs [141]. This SPR sensor chip can be reused in a conventional SPR system without any chemical process for renewal. The use of SPR technology to create new biosensors for virus samples shows promise. Such biosensors are being used to identify RNA viruses such as influenza A/B, SARS-Corona, Ebola virus, Middle East Respiratory Syndrome (MERS), Zika virus and Dengue virus.

#### 4.2.2. Surface-Enhanced Raman Spectroscopy

Surface-enhanced Raman spectroscopy (SERS) is an attractive, highly sensitive and multiplexed assay. Raman reporter molecules modified aptamers are usually immobilized on a nanostructure metal-dielectric substrate to recognize and capture viruses. Negri et al. first proposed a label-free SERS-based platform for detecting influenza virus nucleoproteins [142]. Wang et al. proposed a SERS-based immunoassay with digital microfluidics (DMF) for disease biomarker detection [143]. Magnetic beads that have been coated with antibodies are used as solid supports to capture antigen from the sample, resulting in the formation of a bead-antibody-antigen immunocomplex. Antigens can be detected sensitively by strong SERS signals by labeling immune complexes with SERS tags functionalized with detection antibodies. The quantitative detection of avian influenza virus H5N1 was used to demonstrate the utility of the DMF-SERS method, which has a sensitivity of 74 pg/mL, a detection time of less than 1 h and significantly less reagent consumption (30 μL) than standard ELISA.

#### 4.2.3. Electrochemical Detection Electrochemical Aptamer-Based Techniques

Electrochemistry measures the current response or resistivity change to determine the concentration of a detection target. Idili et al. created an electrochemical aptamer (EAB)-based sensor for measuring SARS-CoV-2 stinger (S) protein in a reagent-free and quantitative manner [144]. The technique is based on signals generated by the binding-induced conformational changes of redox reporter aptamers on the surface of gold electrodes. S proteins in buffer, serum and 50% artificial saliva were measured at the picomolar level. The assay results were also shown to be specific.

### 4.3. Serological Tests

Antibodies produced by the body in response to antigenic stimulation are detected by serological tests. Methods that are commonly used include enzyme-linked immunosorbent assay, chemiluminescence assay (CLIA) and LFIA. Many efforts have been made in recent years to improve diagnostic sensitivity, such as Wang’s development of a selenium nanoparticle-modified SARS-CoV-2 nucleoprotein-based lateral flow immunoassay kit and Devora’s development of a device for simultaneous detection of SARS-CoV-2 RNA and SARS-CoV-2 antibodies [145,146].

### 4.4. Other Methods

For large and heterogeneous samples, conventional mass spectrometry (MS) cannot provide accurate molecular weight measurements. MS has driven the development of the histological revolution, including proteomics, glycomics and metabolomics, with the development of Matrix Assisted Laser Desorption and Ionization (MALDI), particularly electrospray ionization [147]. As a result, mass spectrometry can accurately detect the molecular weight and purity of biological macromolecules such as polypeptides, proteins, nucleic acids and polysaccharides with high accuracy. Virus detection is now being done using liquid chromatography mass spectrometry, electrospray mass spectrometry and Matrix-assisted laser desorption/ionization-time-of-flight mass spectrometry (MALDI-TOF MS) [148,149,150,151,152]. Many assays have been developed on a chip as modern detection techniques become more miniaturized and portable. Mass spectrometry can also be combined with nanoelectromechanical systems to allow for miniaturization, as this technique allows for direct mass measurement without relying on the charge state of the sample [153]. 

### 4.5. Advantages and Limitations of Detection Techniques

In general, nucleic acid assays are sensitive, specific, rapid and accurate and can generally be used as a detection basis. RT-PCR is a mature technology, but it requires strict process control to avoid false negatives or false positives. The LAMP assay is currently thought to be inferior to RT-qPCR in terms of assay sensitivity and specificity, but it has the advantage of a quick turnaround time and ease of setup [154]. For the new crown assay, the commercially available RT-LAMP kit results are consistent with RT-PCR results with a clinical sensitivity of 76.3% [155]. The CRISPR assay’s performance varies, with sensitivity ranging from 80% to 100% and specificity of 100%, and it has great potential for use with POCT devices or high-throughput testing platforms. Although the antigen assay is less sensitive than the nucleic acid assay, it has the advantages of convenience and speed, and it can be used as an individual assay in most cases. The sensitivity of serological methods is related to the duration of onset, with little difference in specificity, which is greater than 95%. It is a powerful diagnostic tool, but it is susceptible to false positive results due to interference from other substances in the blood sample [156].

The first limiting factor is that most current detection systems rely on fluorescence detection, which has high sensitivity, high specificity and real-time monitoring. However, this means that it requires the integration of a number of optical components, excitation sources, photodetectors and filters outside of the device [157]. Another limitation is the requirement for an additional sample amplification step for diseases with low viral load, which requires a thermal module to achieve PCR or isothermal amplification [158,159]. Furthermore, NAAT technologies based on microfluidics lack multiplex capabilities [160,161]. Rigorous validation is still required to translate these technologies from research to practical usage. In addition, the application of molecular diagnostics on microfluidic chips imposes certain requirements on the concentration and purity of samples. In the front-end sampling, the environmental conditions can also have certain effects on the samples: uncentrifuged or unfiltered samples may have large PMs, which may lead to clogging of the microfluidic chip; the chemical composition in different environmental aerosols may affect the pH of the samples; there may also be some activation or inhibitors in the environmental aerosols affecting the enzyme activity in the reaction solution. All these have the potential to affect the reaction in the microfluidic chip.

To summarize, responding to viral disease outbreaks necessitates the use of rapid, accurate and sensitive detection technologies to expedite diagnosis. Microfluidic chips have become a powerful tool for virus molecular analysis over the last 20 years due to inherent advantages such as high integration, accuracy, low reagent consumption and fast response time [162,163]. The ideal microfluidic assay chip should be capable of automating its operation and meeting the goal of “sample input and result output” [164,165,166]. Therefore, the three fundamental steps for conducting experiments on the chip are sample preparation, target amplification and product detection. Despite the large number of papers on nucleic acid amplification and detection that have been published, truly miniaturized chip-scale devices remain elusive in the laboratory and industry, with few examples of fully integrated “sample-to-result” microfluidic devices. Applying microfluidic-based integrated diagnostic devices to CDs could be a promising research topic.

## 5. Application Scenario-Dependent Devices for Airborne Virus Detection

The combination of an aerosol collector and a detector into a single CD reduces the detection time of airborne viruses and speeds up emergency response. The performances of CDs are crucial for the various application scenarios [9,73,74]. As illustrated in Figure 4, the front-end aerosol collector determines the CD’s flow rate and collection efficiency, while the back-end hydrosol detector determines the CD’s sensitivity and other crucial factors, such as response time.

As stated in Section 4.1, there is a quantitative relationship between the concentration of environmental virus aerosols and the concentration of sample. It is formulated as follows.
$$C_{EVA}\cdot CE\cdot F\cdot T=V\cdot C_{HVS}$$

*C_EVA_*: Concentration of environmental virus aerosols

*CE*: Collection efficiency of collector

*F*: Air flow rate

*T*: Sampling time

*V*: Sample volume in detector

*C_HVS_:* Concentration of hydrosol viral sample 

Although this equation can calculate *C_EVA_* and *C_HVS_*, this relation can actually be affected by other environmental factors such as temperature, humidity, ventilation, light conditions, etc.

Various viral aerosol particulate samplers, collectors and detectors have been developed. The primary performance parameters of these devices are collection efficiency, flow rate, sensitivity and response time [167,168]. It is technically difficult to create a device with extreme high performance across the board. As a result, different combinations of principles can be used to improve some of the performance to meet the needs of the application scenarios. As shown in Figure 5, The CDs are categorized into environmental detection device and individual detection device in accordance with various application scenarios. Environmental detection device is further subdivided into environmental monitoring device and environmental detectors.

The first crucial stage is individual detection in the entry. Second, the local environmental virus aerosol can be monitored by using portable environment sensors in the ventilation of busy, complicated locations. In order to continuously track environmental virus aerosol, a high-flow environment aerosol monitoring instrument is used in the mechanical ventilation of the building. When these three situations are combined, a building’s indoor air environment and biosafety can be thoroughly monitored. To develop these three kinds of devices, different primary mechanisms and enhancement techniques need to be used in the development of aerosol collector. Meanwhile, the back-end hydrosol detector should employ appropriate detection techniques. Thus, when using various sample collection mechanisms and detection techniques, scenario-dependent devices are feasible.

For the actual circumstances of various scenarios, there are various options. Thus, it is feasible to combine an environmental monitoring device, environmental detectors and an individual detection device in various forms for a high efficiency detection of airborne viruses. This chapter will provide a building ventilation and airborne virus detection strategy based on the above-mentioned device categories. Aside from the scenarios mentioned above, the main ventilation pipe and sewer pipe in a building both contribute to high levels of indoor air pollution. It is critical to monitor these circumstances using environmental monitoring devices.

### 5.1. Device Classification

#### 5.1.1. Environmental Monitoring Device

The outdoor environment is multi-factorial and complex, and outdoor aerosol monitoring necessitates collaboration with related fields such as environmental monitoring, air pollution control and weather forecasting, as well as the establishment of large monitoring stations for multi-indicator environmental aerosol monitoring and prediction [169]. The CDs mentioned in this review are for monitoring, detection and early warning of viruses in the indoor environment, and their proper application is critical for developing solutions for fighting airborne diseases (Table 1 and Table 2) [170]. Here, we classify indoor environmental monitoring equipment into two groups: high-flow environment aerosol monitoring devices and portable environmental sensor.

##### High-Flow Environment Aerosol Monitoring Device

The volume of aerosols in the living environment is large for indoor environments that require constant ventilation [171]. For example, an office with a floor area of 20 m^2^ and a floor height of 2.5 m contains nearly 50 m^3^ of aerosols. In the environment, viral aerosol PMs have a low concentration and a wide particle size distribution (attached to other PMs) [23,29]. Therefore, aerosol particle collection devices for environmental monitoring are expected to possess the following characteristics in order to meet the requirements:(1)Aerosol collector: high flow rate, high collection efficiency for the full particle size range.(2)Hydrosol collector: high hydrosol flow rate, high sensitivity, short response time.(3)Many researchers have developed such CDs, and environmental monitoring devices should be able to maintain a high level of particle collection efficiency at high flow rate while also delivering hydrosol samples to the back-end high-sensitivity detection module in real time.

##### Portable Environment Sensor

For indoor environments with complex ventilation, it is difficult to monitor inlets and outlets uniformly, or its monitoring cannot meet the actual prevention and control requirements. The development of portable environmental sensors for patrolling or the deployment of indoor environmental aerosol detection can help monitor the indoor environment of the virus aerosol. Such portable environmental sensors should possess the following characteristics: (1)Aerosol collector: high flow rate, high collection efficiency for particle size range of human-generated PM.(2)Hydrosol collector: high sensitivity, short response time.

In addition, this device is expected to be portable and have low energy consumption. Due to energy consumption, portability and other needs, the flow rate of portable environment sensor can be appropriately reduced compared to a high-flow environment aerosol monitoring device.

#### 5.1.2. Individual Detection Device

An individual detection device is a device that collects air directly from a person’s exhalation [172]. The amount of air exhaled by a person in a single breath is about 10 mL/kg, and with an average weight of 70 kg, the amount of aerosol exhaled by a person in a single breath is about 70 mL [173]. PMs contained in human exhaled aerosols range from 0.3 to 10 μm, which requires individual detection devices to have ultra-high sensitivity with very small sample sizes [174]. This requirement brings an extremely high challenge to both the front-end collection technology and the back-end detection technology. In addition, people directly use the device and exhale; without settling, the exhaled PM particle size segment is more stable [175,176]. Therefore, the individual CD should possess the following characteristics: (1)Aerosol collector: low flow rate, high collection efficiency for particle size range of human-generated PM.(2)Hydrosol collector: ultra-high sensitivity, short response time.

The individual detection device should achieve the collection of approximately all the PMs in a limited sample volume and deliver them to the detection module. Ensuring that individuals entering important sites do not produce viral PM has significant implications for localized outbreak prevention and control.

### 5.2. Building Ventilation and Airborne Virus Detection

Airborne virus collection and detection for various indoor environments are achievable using a combination of the various devices mentioned above (Table 1). Common indoor environments ventilation is categorized into three types: mechanical (full ventilation), hybrid ventilation (partial ventilation) and natural (natural ventilation) [177]. Full ventilation means that air exchange through ventilation equipment, no air leakage or air leakage can be ignored. Partial ventilation means, on the basis of ventilation equipment, are also accompanied by windows, doors and other air leaks [178]. Natural ventilation refers to ventilation only through windows, doors and other vents. As illustrated in Figure 5, various types of devices correspond to their appropriate application scenarios.

#### 5.2.1. Critical Sites

The use of nucleic acid testing on all personnel entering a critical site is inefficient, impractical and expensive. As shown in Figure 5A, it is critical to screen people entering a critical site with an individual detection device at the entrance. The strategy will ensure that people entering a critical site do not expel viral PMs. Following the subject’s exhalation into the mouthpiece, the individual detection device detects whether the exhaled breath contains virus particles in a relatively short period of time.

An individual detection device needs to possess ultra-high collection efficiency, and the water-based condenser of PM is particularly suitable for the development of such device. However, a problem is that the energy consumption of such devices is generally high [179]. The miniaturized material of the valley power storage technology for the heating section of the water-based condenser is expected to solve the high energy consumption problem of such devices [180].

#### 5.2.2. Full Ventilation Environment

A full ventilation system with an inlet and an outlet and two ventilation devices is shown in Figure 5B. Because the inlet air volume is slightly greater than the outlet air volume, the system is in a positive pressure state to prevent outdoor air from entering. The inlet and outlet of all ventilation systems require a high flow rate and an efficient sampling device for the indoor environment. Several situations will occur when the CD of the inlet and outlet produce positive or negative results. For a positive inlet and negative outlet, virus aerosol is imported from the outside. For a negative inlet and positive outlet, the infected person is indoors, and viral aerosol is not imported from outside. Both indoor and outdoor viral aerosols have a positive inlet and outlet. For inlet and outlet both being positive, there are both indoor and outdoor viral aerosols. The staff of the Centers for Disease Control (CDC) can respond quickly to the above situations and target different means of prevention and control.

#### 5.2.3. Partial Ventilation Environment

##### Partial Ventilation with Return Air

Shown in Figure 5C, there is local ventilation system with return air and with an inlet and an air returning duct. Inlet ventilators deliver fresh air. Air returning ventilators circulate air, and the system is under positive pressure.

For the indoor environment of partial ventilation, with a return air system, which is the same as the full fresh air system, a high flow rate and high efficiency collection device is used in the inlet and air returning, which can produce a variety of positive/negative results. When the return system’s air volume does not differ significantly from the inlet air volume or when the natural fugitive air volume is low, these two devices work together to achieve better disease control. Inadequate return air volume can cause some of the air to naturally escape. The air returning to the CD does not fully reflect the positive situation of people inside the environment, and sampling every air escaping is impractical. Using the portable environmental sensors with patrolled or distributed detection can compensate for these deficiencies.

##### Partial Ventilation-without Return Air

The local fresh air-no-return air system, as depicted in the figure, has one or more inlets. Gas fugitives occur in gaps such as windows and doors, and the system is under positive pressure. The CD of inlets can still monitor the pollution of the indoor environment by external aerosols for the partial ventilation and no return air system, but the airflow of the indoor environment is more complex and cannot be monitored for each fugitive port. Portable environmental sensors for patrol or distributed detection can improve indoor viral aerosol monitoring.

##### Natural Ventilation

Natural ventilation, as shown in Figure 5D, relies on the flow of gas through windows and doors to achieve air exchange. Detecting all air leaks is inefficient and impractical due to the uncertainty of wind direction. The use of a portable environment sensor has the potential to improve building ventilation and airborne virus detection strategies.

In summary, full ventilation with its well-defined outlets and inlets enables clear monitoring using a high-flow environment aerosol monitoring device. Partial ventilation with well-defined inlets, but with more outlets and air leaks, can be monitored by placing high-flow monitoring equipment in the inlet section, combined with a portable environment sensor. Natural ventilation does not have clear entrances and exits and can only be monitored using portable environmental sensors for distributed or patrol monitoring. 

When developing prevention strategies for each ventilation type of indoor environment, it is important to determine the duration of sampling. The sampling time for individual detection is only one breathing time (3–5 s). Patrol-type detection with intelligent robots will patrol local environments with many and few individuals, so the results should be reported in 30–60 min cycles. High-flow detection equipment fixed in the ventilation port will take longer to accumulate the virus concentration in the liquid sample, so the sampling time should be estimated according to the collection efficiency of the actual equipment, and the sampling time used in the current study is between 1 min and 6 h [23,80,81,82]. Although virus aerosols can be transmitted between rooms due to the presence of sewer stacks and central air conditioning systems, they are most commonly transmitted within rooms. Readers interested in learning more about between-room transmission should consult the following literature, which will not be repeated in this review [181,182,183].

### 5.3. Forward to an Intelligent Detection Strategy of Airborne Virus

Numerous studies show that improving the quality of the indoor environment can reduce viral transmission [184,185,186,187]. To improve indoor air quality, the airborne virus sampling devices can be combined with air purification/filtration techniques and other processes. Airborne virus sampling devices, particularly those with high throughput, enable virus detection to be combined with indoor space ventilation in future building design. Indoor aerosol environmental monitoring requires three types of devices: conventional monitoring devices with certain collection efficiency at high flow rates, high efficiency collection devices with a portable premise at high flow rates and individual monitoring devices with nearly 100% collection efficiency at ultra-low flow rates. The bullets in this section are a comment on how to develop an intelligent strategy for airborne virus detection.
(1)Individual detection devices with ultra-high sensitivity and low flow rates are used to ensure that no viral aerosol PMs are produced by people entering a critical site.(2)A high-flow environment aerosol monitoring device with an ultra-high flow rate is used to collect and monitor viral PMs in the full ventilation environment’s inlet and outlet, as well as the partial ventilation environment’s inlet and return pipe.(3)A portable environmental sensor is used as a supplement because the air leakage from partial and natural ventilation is insufficient for regular monitoring.(4)The collection device at the front end determines CD collection efficiency, but the detection module at the back end determines the specific length of response time, and the detection limit is influenced by both the collection efficiency and the detection method.(5)The enhancement technique improves indoor air quality and contributes to the prevention and control of airborne diseases in indoor environments by lowering PM concentrations and virus activity in inlets and return pipes.

## 6. Conclusions and Future Perspectives

It is crucial for the prevention of airborne transmission diseases to develop various aerosol viral particle collection devices based on the fundamental mechanisms in aerosol science. To deal with the hazards caused by aerosol viruses, such as COVID-19, multiple primary mechanisms enhancement techniques and sensitive detection methods should be used to make aerosol particle collection and detection devices with higher performance. 

Current aerosol particle collection methods vary greatly, and uniform standards are lacking and difficult to establish. For example, almost all of devices mentioned in the above-mentioned literature claim high flow rates. However, the flow rates of at least thousands liter per minutes, according to the comparisons in this review, may be considered as high flow rates. Other than the flow rate, another critical metric is enrichment capacity which indicates the particle concentration of the liquid sample exported by the device. The low-flow devices are still valuable in applications such as individual monitoring and air purification, which require quite high collection efficiency. This specific requirement may be met by particle amplifiers and filters with ultra-high efficiency.

Current epidemic control methods rely on monitoring individuals for nucleic acids and antigens. This means that when we had public social events, the environmental biosecurity of their rooms is marked with a random or mystery box. Only if an infection is present can testing and symptomatic treatment be performed. Indoor ambient air monitoring is becoming increasingly important as viruses continue to mutate and as bacteria and fungi continue to pose a risk. The flow and efficiency of sample and collect will have an impact on the scenario of equipment application, and this paper examines the existing principles and their corresponding scenarios. This paper summarizes the sensitivity, specificity and detection time of the detection techniques themselves, which can provide future research directions.

## Figures and Tables

**Figure 1 ijerph-20-05471-f001:**
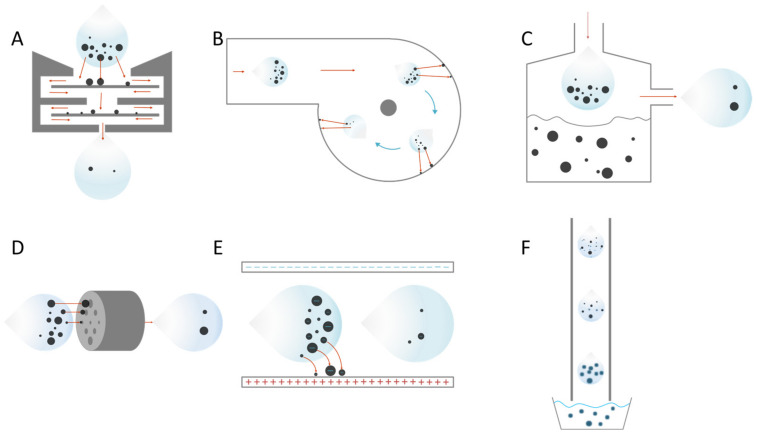
Schematic illustration of primary mechanisms for collection of airborne virus. (**A**) Impactor; (**B**) Cyclone; (**C**) Impinger; (**D**) Filtration; (**E**) Electrostatic precipitator; (**F**) Amplification. Images were modified from [59] with permission.

**Figure 2 ijerph-20-05471-f002:**
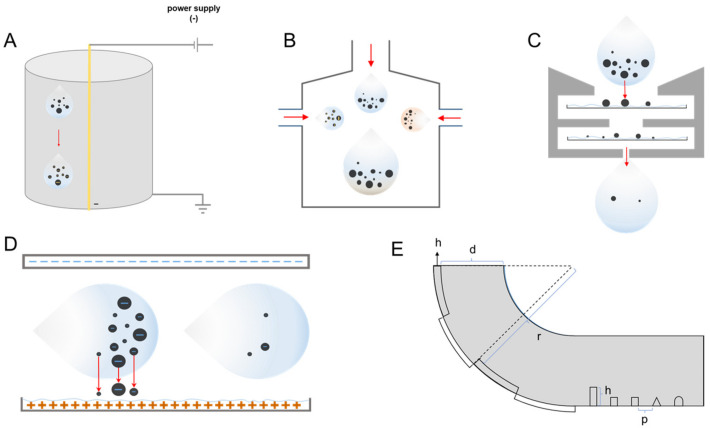
Schematic illustration of enhancement techniques for collection of airborne virus. (**A**) Precharge; (**B**) Premixing; (**C**,**D**) Liquefied collection plate; (**E**) Cavities and Ribs.

**Figure 3 ijerph-20-05471-f003:**
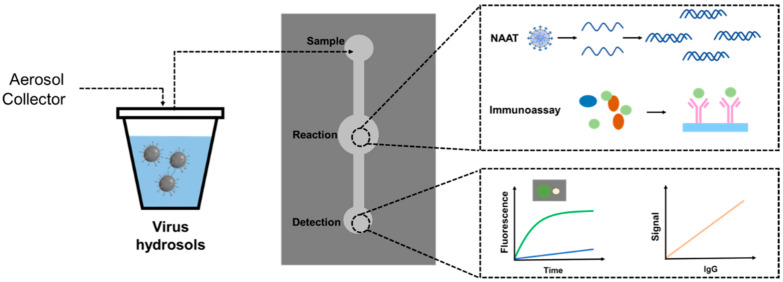
Detection mechanisms of microfluidic devices for airborne virus. NAAT: Nucleic acid amplification test.

**Figure 4 ijerph-20-05471-f004:**
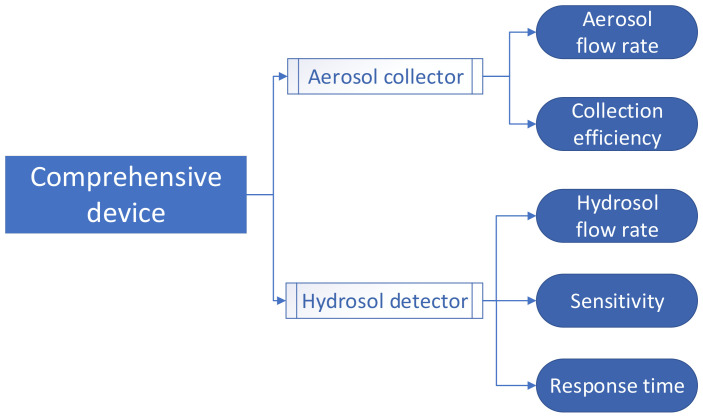
Properties and critical performance parameters determined by the aerosol collector and hydrosol detector in airborne virus detection CDs.

**Figure 5 ijerph-20-05471-f005:**
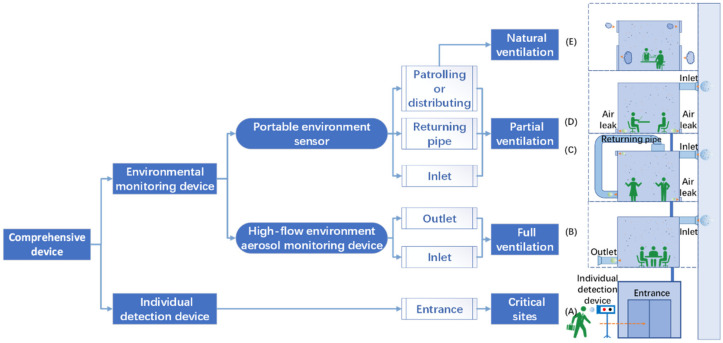
Airborne virus detection strategy for various indoor environments. (**A**) Critical sites in indoor environment; (**B**) Full ventilation environment; (**C**) Partial ventilation with return air; (**D**) Partial ventilation without return air; (**E**) Natural ventilation.

**Table 1 ijerph-20-05471-t001:** Comparisons on the Comprehensive Devices of Airborne Virus Collection.

Device	Flow Rate (L/min)	Collected Particle Size (µm)	Efficiency (%)	Applied Principles	Device Classification	Publications
Highly efficient in-line wet cyclone air sampler	1000	1	50	Cyclone, impactor	Environmental monitoring device	[31,32]
		1.5	78.3			
		>3	about 100			
High-volume sampler for size-selective sampling	1000	Stage 1: >10	50	Cyclone, filtration	Environmental monitoring device	[33,34]
		Stage 2: 2.5–10	<56			
		Stage 3: <2.5	100			
Personal Electrostatic Particle Concentrator (EPC)	1.2	0.05–2	99.3–99.8	Impactor, electrostatic precipitator	Individual detection device	[35]
Exhaled-Breath Bioaerosol Collector (G-II)	130	>0.05	>85	Amplifier and impactor	Portable environment sensor	[36]
Automated Electrostatic Sampler (AES)	1.2	0.3–0.4	>70	Liquefied acquisition plate, precharging, electrostatic precipitator	Individual detection device	[29]
		0.65–0.8	>90			
		0.8–2.0	100			
High air flow-rate electrostatic sampler	40	0.109	88	Liquefied acquisition plate, electrostatic precipitator	Portable environment sensor	[24]
	60		79			
	80		82			
	100		71			
Electrostatic precipitation-based portable low-cost sampler	10	0.01–10	>80	Premixing, electrostatic precipitator	Portable environment sensor	[37]
Integrated microfluidic electrostatic sampler (IMES)	2.8	0.2–10	>90	Liquefied acquisition plate, pre-charging and electrostatic precipitator	Individual detection device, portable environment sensor	[38]
	13.2		>60			
Viral aerosol sampling system using a cooler and steam-jet aerosol collector (SJAC)	12.5	0.03–0.1	70–99	Cyclone, aerosol premixing	Portable environment sensor	[39]
Low-Cost Micro-Channel Aerosol Collector	1.5	0.5	50	Micro-Channel cyclone	Individual detection device	[40]
		>1	90			
		>2	100			

**Table 2 ijerph-20-05471-t002:** Summary of potential detection techniques for airborne virus.

Detection Techniques (Acronym)	Detection Techniques (Full Name)	Description	Advantages	Disadvantages	Publications
PCR	Polymerase chain reaction	Using primers, thermal cycling, thermal cycling needs to go through three temperature changes.	High accuracy and specificity, low detection limit, suitable for all kinds of samples.	Need a long time and special laboratory environment.	[11,41]
LAMP	Loop-mediated isothermal amplification	In vitro amplification of nucleic acids at a constant temperature of typically 60–65 °C.	Cheap, just a hot plate.	High demand for primers, false positive rate may be high.	[42,43,44,45,46]
CRISPR	Clustered regularly interspaced short palindromic repeats	Cas enzyme indiscriminately cuts the surrounding single strand after activation.	Fast and specific.	Relies on the preamplification to detect the targets when concentrations below femtomolar level.	[47,48,49,50,51,52]
ELISA	Enzyme linked immunosorbent assay	An enzyme is used to display and convert readings in a measurable manner based on the interaction between antigen and antibody.	The detection takes only ten minutes, no special instrument required.	Sensitivity and specificity are limited. Does not apply to all virus types.	[41,53,54]
SPR	Surface plasmon resonance	Light hits a sensor which covered with a metal film. Measures the intensity of the refracted light.	The analyte concentrations can be determined and data on biological reaction kinetics can be obtained, with a low LOD character.	Sensors are difficult to reuse.	[55,56]
MS	Mass spectrometry	Mass spectrometry and PCR technology are perfectly combined in nucleic acid mass spectrometry, making it ideal for the investigation of dozens to hundreds of gene loci.	It can be used for direct sample detection while the high-throughput features support multi-site multi-targeted detection.	Professional needs are comparatively high, standardization plans are few, and automation plans are expensive.	[57,58]

## Data Availability

Not applicable.

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
