# Peer review of "Mechanisms, Techniques and Devices of Airborne Virus Detection: A Review"

_ijerph, 2023, doi:10.3390/ijerph20085471_

Round 1

Reviewer 1 Report

This is a broad review of the techniques and instruments used to detect air-borne concentrations of viruses. The fundamentals of any technique are to collect aerosol into a liquid bath and then assay the hydrosol for the concentration of bioaerosols. The assay devices have various sophistications allowing further speciation of the bioaerosols. The authors review size ranges, efficiencies, limitations, and other aspects of the various instruments used to complete this three step measurements process.

My expertise is limited to aerosol measurement and collection, so I have a limited basis for reviewing this paper. The authors description of the various aerosol collectors is not the best and I have a few specific comments below about some of the language used, but for the purposes of this paper the descriptions mostly suffice. Such comments, and often they were highlighting difficulties with the English, were stopped after page 6, as I only skimmed the paper from then on.

The paper is quite tedious to read with no real scientific contribution, and would only be useful to someone directly involved with these measurements who wished to perhaps compare several methods. All the figures are limited to drawings to illustrate the elements of a measurement.

The paper could perhaps be improved with efficient use of tables to compare the various aspects of the three measurement fundamentals, but the tables should be carefully and clearly constructed so the reader could easily glance across a row and down a column to compare the various instruments/methods. The one table included is not well constructed. Only columns 2 and 3 provide quantitative information and each number also includes the units. The units need to go at the top in the column header, e.g. L/min, µm, %. Size and efficiencies should be separate columns. All the sizes should use the same units. There must also be other quantitative characteristics of the various methods which could be similarly compared. Here the rows are labelled with a literature title, not an instrument type. Such table would eliminate or at least limit the endless paragraphs citing such numbers later in the text, e.g. sections 3.2.5, 3.3.1, 3.3.2. Perhaps something similar could be constructed for all the various virus types which can be detected.

While the English is mostly quite good, there are still some difficulties as mentioned above and called out in a few places below. No line numbers are provided so I indicate the location with a direct quote from the manuscript divided by page numbers.

Page 1  …capable of separating and purifying particles from aerosols… This is confusing terminology since particles and aerosols are the same thing. What do the authors intend? That bio-aerosols are separated from the particles on which they are carried? What is being purified?

For example, quantitative real-time polymerase chain reaction (qRT-PCR), which is a common method for virus detection due to its high sensitivity and reliability.” Is not a complete sentence.

The first step is to collect the particles from aerosol to hydrosol” I think the authors mean convert the particles from aerosol to hydrosol, or more simply collect the aerosol in a liquid bath.

Most of Figure 1 is wasted space, i.e. the people, table, cloud. Only the blue box is called out and even this simple concept hardly needs a figure.

While combining bioaerosol samplers with other analytical detection techniques to achieve real-time monitoring of airborne viruses is challenging [19].” This a clause not a sentence. Remove the while and it becomes a sentence.

Page 2

“Inertance” has a very specific meaning, and it is not what the authors are using it for.

3.1.1 “The inertia of the particles in aerosol is greater than that of the rest of the aerosol components.” This only becomes important if there is a change in the aerosol flow. The authors’ few sentences in this section are not helpful to explain the phenomena.

“High flow rate collection is not possible when the particle size is at the nano or micron level since it only selects the lower limit, not the upper limit of the particle size.” Confusing. What only selects the lower limit? The flow rate or the filter or both? Does a filter have an upper limit? If not why mention one. I think filters generally collect all particles above a certain size.

Page 3

3.1.3 Electrostatic precipitators are usually paired with an electron source to cause the aerosol to assume a Boltzmann equilibrium charge distribution. Then the precipitator can be pulsed positive / negative to capture all the charged particles. Most do not rely on just the ambient particle charge distribution.

3.1.4 The amplifier does not cause the particles to combine with other particles. Rather that high humidity causes each particle to swell with the water which condenses on the particles. Thus each particle gets larger due to deliquescence/condensation and thus grows and becomes easier to collect, not because the particle are coagulating. The figure 2F illustrating this is not correct.

Page 4

In nature, viral particles have relatively weak negative charges.” What is the reason for this? How has it been demonstrated? Where is the reference?

3.2.2 What are the special features? This section varies from procedures to enhance collection to procedures to mitigate the viral load of an aerosol population, quite different goals.

Page 6

“Generally, the concentration of hydrosol samples after collection by aerosol collector insufficient.” English needs fixed. Add a verb.

Reviewer 2 Report

Mechanisms, Techniques and Devises of Airborne Virus Detection: A Review

Chang et al. 

Helpful review of lots of techniques and examples of bioaerosol sampling applications.

Could improve clarity throughout and pay attention to a few main variables such as sampling time, limits of detection on downstream assays, impacts on culturability, expectations for nucleic acid detection given what is known about how much virus is shed into exhaled breath and how much air is being sampled and the ventilation in the space.

To the extent possible, highlighting the advantages of samplers in practice that have detected SARS-CoV-2 and influenza and the conditions under which they were used, could add something to the review. 

Thank you for your contribution!

Introduction

As written, it sounds like you are defining airborne transmission as that which occurs at long-range (“greater than 1 to 2 m from the infected person”). Please clarify that while long-range transmission is possible and a notable feature of respiratory viruses due to the ability to superspread within a room, that the risk of airborne transmission at close-range is generally always higher than that of long-range transmission. 

Consider citing paper by Tang, Marr, and Milton (10.1016/j.jhin.2021.05.007). 

It sounds like airborne transmission has not played a role in other important epidemics/pandemics. But it is widely accepted that airborne transmission mode is an important mode for respiratory viruses and likely drives community spread. 

Consider citing Tang, Tellier, Li (10.1111/ina.12937). 

Consider citing Milton Rosetta Stone Paper (10.1093/jpids/piaa079) — importantly, the mode of exposure of viral aerosols is via inhalation.

Consider citing the wealth of studies by Donald K. Milton’s group that quantified viral load of people with influenza viruses and, SARS-CoV-2. We know that people she lots of virus into their exhaled breath and do not need to be coughing to do so. There is great variation in exhaled breath viral shedding and temporal dynamics over the course of infection. Work by Kristen Coleman and others shows that coughing and singing can increase the viral load emitted into aerosols from the respiratory tract. 

The sentence that cites reference #12 seems to conflate physical and biological decay of the viral aerosols. Viral infectivity can remain for many hours (beyond 8 — see Fears et al (10.3201/eid2609.201806).

For a reference that describes well the paradigm of airborne transmission and decay mechanisms — please see Roy and Milton (10.1056/nejmp048051)

The word “concomitants” should perhaps be “contaminants”?

The phrase “pathogenic PMs in bioaerosols” doesn’t seem to make much sense. Could simply say “potential infectious bioaerosls”?

Aerosols are simply particles in the air. Please review the language in the introduction to be clear. “Separating and purifying particles from aerosols” doesn’t make sense to me. Are you trying to say “isolating and concentration virus particles from aerosol samples?” Perhaps something like that would be more clear?

The fraction ≤5 µm is important because this is the only size that can reach the intrathoracic space following inhalation exposure (only size that can transmit TB, for example, and important for influenza and SARS-CoV-2 transmission). Adding some additional size fractions below this and above this would be helpful. Considerations include the modes of exposure reviewed in Milton Rosetta Stone paper and also those related to the transportation of the virus in the air (e.g., see Bueno de Mesquita et al. 10.1111/ina.12965 figure 1) or other analyses from Yuguo Li.

Consider making the goals of the review (specific aims, for example) more explicit. What exactly are you reviewing, based on the gaps identified in the hypothesis?

Methods

Why no NIOSH bioaerosol sampler or Thermo AerosolSense sampler included in the table 1?

Consider addition colloquial sampler names if possible to Table 1

What are the different options for “Device classification” in Table 2 and what do they mean? What does “individual detection device” mean, for example?

Can PR differentiate different types of airborne viruses?

Fig 2B. Wondering if the arrows pointing to the particles settling on the sides should be moving in the direction of the air flow? 

Fig 2. General comment — consider describing the meaning of the blue drops that are filled with particles. Is the blue drop referring to the air that is being sampled (I presume)? 

You could simply show the particles (aerosols) of different sizes? The blue drop sort of looks like it is an aerosol that contains a number of different viruses, etc. Removing the blue “drops” seems like it might improve clarity of this figure.

Sampling Techniques and Devices of Airborne Viruses

3.1.1. Are you saying that viruses contained within the aerosol particles come out, e.g. the more solid components of the droplet nuclei? I think that the entire aerosol particle would remain intact. 

I think this is because I’m talking about aerosol particles as Wang et al (10.1126/science.abd9149) fig 2 describes them. The entire droplet is the aerosol particle.

Consider talking about bioaerosols as “bioaerosols” and not as PM.

Consider removing the “COMMENTS” note within the 3.1 subsections?

Consider adding some detail about the advantages and disadvantages of the different sampling mechanisms and some examples of the samplers using these different strategies in practice for influenza or SARS-CoV-2?

For example, what are some of the size cutoffs used for impactors in section 3.1.1.1?

Andersen cascade sampler would be good example?

What do you mean by “smaller particles sizes” that “require extremely high velocities”? Please consider noting specific about sizes and velocities and/or use examples.

Consider increasing the level of detail and/or citing examples of specific samplers for the other classes of samplers?

Consider adding to comparisons of capture efficiencies between sampler types and discussing the considerations for downstream analyses - for example, some interesting discussion is contained in https://patents.google.com/patent/US10502665B2/en regarding exhaedl breath bioaerosol sampling.

There seem to be problems with recovery of sample from filter for downstream PCR/culture. Consider discussing this. 

OK, I see there is a section later on this.

3.2.2.

Would be interesting to hear more about the potential for TTO on SARS-CoV-2 biological degradation. Same for acidic hypochlorite water. Perhaps a bit of discussion on the mechanisms of action (even if only hypothesized, and stated as such if so) would be useful?

Bringing it back to how premixing influences aspects of collection and utility for downstream analyses could be further developed?

3.2.3.

I think you are saying that there is greater potential for viral infectious decay on dray plates (when stating “activity of VPs on a dry plate cannot be guaranteed”) but also this is also a challenge with most methods.

3.2.4.

What do you mean by “smaller particle size”? Is it completely linear? Perhaps speak more directly to the study cited and the particle sizes measured?

What is p/e?

Perhaps mention methods for recovery of samples with the cavities and ribs?

What shapes of ribs were tried?

3.2.5.

Consider describing HTH in terms of 1) sample concentration, and 2) sample detection, since it sounds like these are two distinct categories/mechanisms?

3.3.1.

May wish to talk about McDevitt et al as collecting exhaled breath aerosols. I suppose this could be equivalent to PM collection, but is more specific to the particles that are coming from the respiratory tract and being grown up through the addition of water to the system, before being condensed into liquid drops.

4.1.

Consider discussing some of the considerations about detection and quantification limits on nucleic acid detection techniques. Microfluidics could offer a helpful approach to real-time detection capability, but would need to be concentrated with enough sample to have a reasonable chance of detecting something.

For example, how much sampling time might be envisioned before moving a sample into a downstream analysis and how much room air would be needed in a sample, given the sampling efficiencies for particle sizes of interest, and given some knowledge of how much virus is shed into exhaled breath, in order to enable detection?

4.2.

Perhaps mention something about how SPRs differentiate between pathogens?

4.5.

Nice section here on advantages and limitations

Interference may occur not only in blood samples, but also in upper respiratory swab samples. You might speculate about the potential for interference occurring for aerosol samples, that will have a different composition.

5. 

Figure 5. Perhaps a little figure showing the aerosol and hydrofoil detectors with respect to the flow rates, efficiency/sensitivity, and response time would help clarify the main point of the figure?

Figure 6. Would help to have a little more explanation about what is meant in the description of the relationship between ventilation and the sensors. Could improve description in 5.2 to help? For example is the consideration of the ventilation rate (or use of air cleaners or disinfection by UV) having to do with better ascertaining the source generation strength of a bioaerosol, for example (e.g., influenza A bioaerosol shed from an infected person)

5.1.2. 

Good point about the size fractions of bioaerosols from humans. Could situate this earlier on to orient the reader that these sizes are important and also they are shrinking due to evaporation after leaving the supersaturated respiratory tract.

Consider changing viral PMs to viral aerosols since this is normally how the field refers to it.

5.2.2.

OK, it is making more sense what Figure 6 is about having read 5.2.2. Consider moving this up or situating the reader with an example that has actionable public health practice implications.

Although noting that it is rare that relevant bioaerosols would be transported between rooms — the most relevant exposures are mostly occurring within room — and this is certainly where the concentration is highest.

What about placing bioaerosol samplers throughout the room, in addition to, or in place of an inlet? Could have some interesting experiments to learn more about how air is mixing and, critically, moving from one person’s exhalation to another’s inhalation.

Could you say something about sampling times to accompany the suggestions for sampler placement in these different types of ventilated settings?

Could you speak to culture methods and the impact of sampling strategies on the potential for downstream cultivability of viruses or bacteria in the air?
